# Fighting Oxidative Stress with Sulfur: Hydrogen Sulfide in the Renal and Cardiovascular Systems

**DOI:** 10.3390/antiox10030373

**Published:** 2021-03-02

**Authors:** Joshua J. Scammahorn, Isabel T. N. Nguyen, Eelke M. Bos, Harry Van Goor, Jaap A. Joles

**Affiliations:** 1Department of Nephrology & Hypertension, University Medical Center Utrecht, 3508 GA Utrecht, The Netherlands; j.j.scammahorn@umcutrecht.nl (J.J.S.); T.N.Nguyen-4@umcutrecht.nl (I.T.N.N.); J.A.Joles@umcutrecht.nl (J.A.J.); 2Department of Neurosurgery, Erasmus Medical Center Rotterdam, 3015 CN Rotterdam, The Netherlands; e.bos@erasmusmc.nl; 3Department of Pathology and Medical Biology, University Medical Center Groningen and University of Groningen, 9713 GZ Groningen, The Netherlands

**Keywords:** hydrogen sulfide, reactive oxygen species, H_2_S donors, cardiorenal syndrome, thiosulfate

## Abstract

Hydrogen sulfide (H_2_S) is an essential gaseous signaling molecule. Research on its role in physiological and pathophysiological processes has greatly expanded. Endogenous enzymatic production through the transsulfuration and cysteine catabolism pathways can occur in the kidneys and blood vessels. Furthermore, non-enzymatic pathways are present throughout the body. In the renal and cardiovascular system, H_2_S plays an important role in maintaining the redox status at safe levels by promoting scavenging of reactive oxygen species (ROS). H_2_S also modifies cysteine residues on key signaling molecules such as keap1/Nrf2, NFκB, and HIF-1α, thereby promoting anti-oxidant mechanisms. Depletion of H_2_S is implicated in many age-related and cardiorenal diseases, all having oxidative stress as a major contributor. Current research suggests potential for H_2_S-based therapies, however, therapeutic interventions have been limited to studies in animal models. Beyond H_2_S use as direct treatment, it could improve procedures such as transplantation, stem cell therapy, and the safety and efficacy of drugs including NSAIDs and ACE inhibitors. All in all, H_2_S is a prime subject for further research with potential for clinical use.

## 1. Introduction

Hydrogen sulfide (H_2_S) was first described as a pungent toxic gas in “De Morbis Artificum Diatriba”, Bernardino Ramazzini’s treatise on worker’s diseases [1]. Today, we have come to better understand the toxic effect of H_2_S and have determined that this is most likely the result of cytochrome C oxidase (mitochondrial Complex IV), monoamine oxidase, -and/or Na^+^-K^+^-ATPase inhibition [1,2,3]. A brief overview of H_2_S toxicity is given in Table 1. In short, high concentrations disrupt the oxidative metabolism process of cells leading to impairment of organs most reliant on these processes [1]. During research on its toxicity, several groups reported the presence of H_2_S in tissues of healthy humans and laboratory animals [4,5,6]. Although the transsulfuration pathway was described in mammals more than fifty years ago, it was not until 1996 that the role of H_2_S as an endogenous gaseous signaling molecule was proposed [7,8]. Since then, research has expanded on the topic, revealing its role in many physiological processes including those of the renal and cardiovascular system [9,10].

In the body, the kidneys are believed to be the third largest producer of H_2_S, after the liver and gut [9]. All of the known pathways leading to the production of H_2_S have been described in the kidneys. Furthermore, renal homeostasis appears to be under control of H_2_S to some degree, with H_2_S levels contributing to the glomerular filtration rate (GFR), Na+ excretion, and K+ excretion [11]. Various effects mediated by H_2_S have been observed with consequences in both the renal and cardiovascular systems including epigenetic regulation of apoptosis, immunoregulatory effects, cellular protein homeostasis, and its role as an oxygen sensor and/or inducing hypometabolism in cells [12]. These have been extensively reviewed elsewhere [9,12]. H_2_S has also been shown to have powerful antioxidant properties. Indeed, it is part of the reactive species interactome, as defined by Cortese-Krott et al. as the chemical interactions between reactive oxygen, nitrate and sulfur species (resp. ROS, RNS and RSS) and their downstream targets [13,14,15]. In general, H_2_S has shown potential as a biomarker for disease; the reduction of H_2_S levels correlates with renal/cardiovascular disease progression and general mortality [16,17].

The complex interplay between the cardiovascular and renal system leads to pathologies affecting one having consequences for the other [18,19]}. This can lead to new diseases developing or worsening existing ones [20]. This has led to the classification system based on the syndrome created by these interactions, known as the cardiorenal syndrome (CRS) [19,21]. A major component of this syndrome is related to blood pressure control, in which renal renin–angiotensin–aldosterone system (RAAS) activation, sympathetic nervous system, and the pump function of the heart can fall into a vicious circle [22]. However, this does not fully explain the CRS with chronic inflammation, persistent RAAS activation, and ROS signaling being further implicated [20]. Notably, H_2_S has been shown to affect various aspects of cardiorenal interactions [9,10]. Furthermore, H_2_S can be beneficial in the individual pathologies that fall under CRS: heart failure, cardiac hypertrophy, ischemia, chronic kidney disease (CKD), angiotensin related hypertension, and diabetes related renal pathology to name a few [23,24,25]. The antioxidant effects of H_2_S specifically in renal-cardiovascular systems might hold the key to a better understanding or treatment of CRS [11,26,27,28]. Considering the potential role of oxidative stress in this syndrome, it is important to further explore this in a dedicated review.

## 2. Endogenous Production of Hydrogen Sulfide

There are four known pathways resulting in the production of H_2_S in mammals, with the kidneys showing notable activity in all three enzymatic ones [9,29]. Blood vessels also show the activity of these enzymes [30]. Two enzymatic pathways, that of cystathionine-β-synthase (CBS) and cystathionine gamma-lyase (CSE), are grouped together under the broader transsulfuration pathway, discussed in Section 2.1. A third enzymatic pathway exists primarily in the mitochondria, the cysteine catabolic pathway mediated by 3-mercaptopyruvate sulfurtransferase (3-MST), and is expanded upon in Section 2.2. Another non-enzymatic pathway leading to the production of H_2_S has been explored recently and is covered in Section 2.3. The presence of H_2_S throughout the body is thought to be regulated by the hypothalamic–pituitary axis (at least in mice) and diet [31,32].

### 2.1. Transsulfuration Pathway

The transsulfuration pathway was described in mammals during the mid-twentieth century, later rising to prominence due to the unravelling of H_2_S’s potential as a signaling molecule [1]. The result of this pathway is the biosynthesis of L-cysteine from homocysteine, as shown in Figure 1, a process central in the metabolism of sulfur and regulation of cellular redox [33,34]. The enzymes CBS and CSE are essential to this pathway, being able to produce H_2_S independently as well as in concert with each other. CBS, present in the kidney, but less so in the heart, uses a combination of cysteine and homocysteine to generate H_2_S [29]. It can also produce cystathionine from serine and homocysteine. CSE, notably active in the kidney and vasculature, but not the heart (in mice), takes cystationine created from CBS to produce cysteine, which it can also use to generate H_2_S [35,36]. It is important to note that CSE expression is induced by endoplasmic reticulum stress and oxidative stress among other stimuli, whereas CBS is inhibited by the other gaseous signaling molecules, NO, and carbon monoxide (CO). For more details on transsulfuration pathway and its regulation, see Sbodio et al. [33].

### 2.2. Cysteine Catabolism Pathway

Cysteine can be used to generate H_2_S via the cysteine catabolic pathway. D-cysteine is processed by D-amino acid oxidase (DOA) in the peroxisomes to produce 3-mercaptopyruvate (3MP), while L-cysteine and α-ketobutyrate produced from the transsulfuration pathway are turned into 3MP by cysteine aminotransferase (CAT) in the mitochondria, as shown in Figure 2 [37].

The enzyme 3-MST then comes into play, producing sulfides and polysulfides including H_2_S. In turn, H_2_S can reduce these various products. 3-MST is also capable of converting H_2_S into a hydrogen polysulfide. When 3MP is not present in sufficient concentrations for the 3-MST activity, antioxidant cysteine and glutathione concentrations drop, suggesting that these are consumed [37].

### 2.3. Nonenzymatic Pathways

Recently, non-enzymatic production of H_2_S has also been described in blood and in vitro [35]. This process appears to require the presence of iron (Fe^3+^ form) and vitamin B6 in blood. Interestingly, the enzymatic pathway is most prominent in the liver and the kidney, while non-enzymatic production plays a greater role in other tissues [35]. The optimal substrate for this form of production appears to be cysteine, regardless of the D/L- isomer [35]. How this process relates to the redox aspect of H_2_S remains to be explored. Besides this novel pathway, various molecules found naturally in the body can donate H_2_S including thiocysteine, thiosulfate, and polysulfides [38]. Such molecules therefore provide natural leads for therapeutic drugs.

## 3. Antioxidant Mechanisms of Hydrogen Sulfide

Cardiovascular and renal research on H_2_S in rodents indicate that it has the potential to modulate oxidative stress at the tissue and organ level [34]. At the cellular level, H_2_S has been shown to influence cellular redox via four mechanisms [39]. The first is the scavenging of ROS by induction of major antioxidants [27]. Second, cysteine residues in proteins can be modulated by H_2_S, resulting in persulfidation, which, in combination with thioredoxin, potentially protects proteins from oxidative stress [40,41]. Third, H_2_S plays a role in the mitochondria and oxidative respiration production of adenosine triphosphate (ATP) [42,43,44]. Finally, H_2_S can react with metals including iron in the heme of cytochrome c oxidase [3]. Xie et al. have published an extensive review on the topic of H_2_S and cellular redox [27]. Beyond these interactions, it has become clear that there is an interplay with NO, another gaseous signaling molecule [45].

### 3.1. Reactive Oxygen Species Scavenging

Downstream of the transsulfuration pathway, cysteine also acts as an important antioxidant and can be used to produce the major antioxidant glutathione [33]. Glutathione is a thiol produced by combining L-glutamate and L-cysteine by glutamate cysteine ligase and then combining the product (γ-glutamyl-L-cysteine) with L-glycine by glutathione synthase. H_2_S, cysteine, and glutathione can all scavenge ROS by forming disulfide bonds from their –SH residue [33]. However, H_2_S’s role in direct scavenging is thought to be limited, as the concentrations present in vivo are too low for that to be its primary mode of antioxidant activity. Thus, antioxidant effects related to CBS and CSE are not necessarily directly related to H_2_S production, but to other products resulting from the transsulfuration pathway. It is more likely that effects seen at the cellular level are a result of H_2_S’s signaling capabilities. Indeed, the production of glutathione is regulated by known targets of H_2_S signaling including the Keap1/Nrf2 pathway.

### 3.2. Protein Modification

H_2_S has the ability to modify many proteins including Keap1, NFκb, and HIF-1α. One of the major antioxidant pathways is the Keap1/Nrf2 pathway, a simplified version of which can be found in Figure 3. H_2_S has been shown to modulate Nrf2 through sulfuration of Keap1, leading to activation of Nrf2 [27]. In this way, H_2_S contributes to protecting the cell from oxidative stress related injury [46]. Nrf2 in turn regulates the production of major ROS scavengers such as glutathione and thioredoxin, as discussed previously [27,47]. Keap1 also leads to transcription of various antioxidant enzymes such as superoxide dismutase, catalase, and glutathione peroxidase [27]. These aspects of Nrf2 signaling are important for preventing oxidative stress induced senescence [12]. H_2_S also activates NFκb, a cornerstone of the inflammatory pathways activated by ROS signaling, as summarized in Figure 3 [10,30]. NFκb leads to transcription of some of the same antioxidant enzymes that are stimulated by Keap1 [27]. HIF-1α is a third important signaling molecule that is potentially activated or downregulated by H_2_S [48,49]. The signaling pathway of HIF-1α is briefly presented in Figure 3. The regulation is H_2_S-dose dependent, as lower doses appear to upregulate and stabilize HIF-1α while a higher dose downregulates and destabilizes it [48,50].

### 3.3. Mitochondria and Respiratory Oxidation

The major cellular production of ROS is due to oxidative respiration and therefore located in the mitochondria. H_2_S is an essential molecule for mitochondria and regulates the amounts of ROS produced [43]. ROS are in turn important for regulating adaptation in order to promote tissue survival, however, this comes at the cost of proper function when ROS induces oxidative stress [51]. There is a particularly interesting effect in tissues that express the enzyme CSE and/or CBS. This is due to translocation of these enzymes to the mitochondria in response to hypoxia and forced calcium release to the cytoplasm, at least in vitro [43,52]. Translocation of CSE under the influence of calcium increases ATP production under normoxic and hypoxic conditions at the cost of cysteine. However, addition of extra H_2_S through donors leads to decreased ATP production in normoxic conditions [43]. Recently, normoxic perfusion of H_2_S donors in whole ex vivo porcine kidneys resulted in renal oxygen consumption being reduced by over 60% with corresponding decreases in mitochondrial activity [53]. This would suggest that ROS production is reduced by H_2_S through hypometabolism.

Despite the hypometabolism induced by H_2_S, ATP levels, renal function, and histological structure were unaffected, providing evidence that H_2_S can partially substitute oxygen in ATP production, thus reducing the amount of ROS generated by the mitochondria [53]. In the mitochondria, H_2_S interacts with the heme group of cytochrome c and the metal cofactors of cytochrome c oxidase. Located in the intermembrane compartment, cytochrome c transfers electrons between cytochrome c reductase (complex III) and cytochrome c oxidase (IV). H_2_S is able to donate electrons in the mitochondrial ATP production machinery through its interaction with cytochrome c as well as reducing complex IV directly without interacting with complex III [54]. Beyond the implications for ROS production, cytochrome c plays an important role in inducing apoptosis in which H_2_S intervenes.

## 4. Hydrogen Sulfide in Cardiovascular and Renal Physiology

Considering the interactions found between H_2_S and ROS, it is important to note the role of the two in maintaining cellular homeostasis. ROS is a necessary signaling molecule that is maintained at an optimal concentration under physiological circumstances, with too little or too much being potentially problematic [55]. For an overview of the consequences of reduced ROS, see the reviews by Sies et al. [51,55]. Increased ROS at oxidative stress levels forms an essential part of our understanding of the biological role of H_2_S. Under physiological circumstances in the cardiovascular system, H_2_S interacts with the balance between NO and ROS [56]. Renal H_2_S, on the other hand, seems to be under the influence of higher enzyme concentrations [9,29]. In both cases, cell proliferation and functions are influenced by the degree of sulfuration versus oxidation of various proteins [9,29,56]. In this section, we focus on how these cellular and signaling mechanisms take place in the cardiovascular and renal systems and what effects result at the tissue and organ level under physiological circumstances.

### 4.1. Cardiovascular

The production of ROS, in particular hydrogen peroxide (H_2_O_2_), in the heart is important for its ability to adapt to environmental stress [56]. In cardiomyocytes, stimulation of the alpha-adrenergic receptors by noradrenaline leads to the production of ROS through NAPDH oxidase (NOX) [56]. ROS can then in turn oxidize cysteine residues in important signaling pathways such as NFκb and Nrf2. The cysteine residues are sulfurated by H_2_S, providing fine control over these activations considering that the effects of oxidation are dependent on which residues are oxidized [57]. NFκb, a pro-inflammatory pathway, is also activated by angiotensin II (ANG II) and/or mechanical stretch via NOX and in turn ROS can regulate NOX activation through oxidation [30]. This, combined with the ability of H_2_S to activate these pathways without the inflammatory effect, would suggest that the role of H_2_S is in part to limit ROS signaling to physiological levels.

At the tissue level, H_2_S protects against dysfunction through cellular senescence and apoptosis while allowing for adaptation in the form of controlled inflammation, angiogenesis, and proliferation enacted by physiological levels of ROS [49,58]. This is supported by aging studies in rats where the levels of H_2_S were followed. Aged hearts in general are more prone to disease than younger hearts and tend to have lower levels of H_2_S [59]. Furthermore, when looking solely at aged hearts, those with lower levels of H_2_S are more prone to age related pathologies than those that have retained more H_2_S [60]. One of the major driving mechanisms of these types of pathologies appears to be oxidative stress, the state in which ROS supersedes the safe adaptive range [12]. Going a step further, intervening with H_2_S treatment in hearts undergoing oxidative stress can restore redox balance [61].

When it comes to the vasculature, ROS are well established signaling molecules under physiological circumstances in the blood vessels, with different ROS having different properties in how they distribute in and out of the cell [55]. In the long-term, H_2_S prevents and reverses vascular remodeling through preventing smooth muscle proliferation and apoptosis [55]. H_2_S also interacts with NO for more rapid changes such as control of the vessel diameter, particularly when NO is depleted [62,63]. H_2_O_2_ and NO can cause vasodilation, whereas ROS other than H_2_O_2_ such as superoxide (O_2_^−^) causes vasoconstriction [56]. Overall, H_2_S appears to be a vasodilator [30,63]. However, the interactions are much more complex than their individual effects. Indeed, stimulation of the angiotensin receptor I or endothelin (ET) receptor A leads to NADPH production of ROS and vasoconstriction while ET receptor B causes vasodilation [62]. It may be possible that H_2_S regulation has the strongest effect on the dominant ROS being produced (be that O_2_^−^ or H_2_O_2_), resulting in an opposite effect on the vessels.

### 4.2. Renal

The kidneys are a major producer of H_2_S, as indicated by their expression of CBS, CSE, and 3-MST [9,11]. H_2_S causes similar effects to those of the cardiovascular system due to the interactions with the same pathways to protect against inflammation and apoptosis by regulating ROS signaling [9,64]. Likewise, the defenses that H_2_S provides against oxidative stress are important for maintaining cellular function in the kidneys. Furthermore, renal H_2_S is also reduced due to the effects of aging in the kidneys [65]. One of the mechanisms behind this effect is the modification of p21 regulated senescence [65]. On the matter of NO–ROS interactions, there is indication that H_2_S works in tandem with NO in the kidneys by upregulating endothelial NO synthase and thus stimulating NO production [9]. However, there are also effects that are unique to the kidneys.

H_2_S is involved in renal homeostasis by increasing GFR and the excretion of Na+ and K+ through inhibition of the Na^+^–K^+^–2Cl^-^-cotransporter and Na^+^–K^+^–ATPase [11]. This explains part of H_2_S’s ability to regulate blood pressure, in combination with its role in the RAAS system through cyclic adenosine monophosphate (cAMP) regulated renin release and its aforementioned regulation of blood vessels [66,67,68]. However, research has shown that H_2_S is metabolized in the presence of oxygen (O_2)_ in renal tissue [69]. In this sense, it acts as a sensor for O_2_, becoming more active under hypoxic conditions and enhancing renal blood flow to alleviate hypoxia in the kidneys [9,11]. Hypoxia also leads to systemic signaling from the kidney to increase the number of red blood cells through erythropoietin (EPO) [70]. In other words, under physiological circumstances, there is a balance between the gases H_2_S, O_2_, and NO that help maintain hemopoiesis and renal homeostasis [64,71].

## 5. Role in Cardiorenal Syndrome Pathologies

H_2_S production levels have been implicated as potential disease markers in various pathologies [38]. Furthermore, reduced sulfate excretion has been shown to be a potential biomarker for renal and cardiovascular diseases as well as overall mortality in the general population [16]. When examining renal and cardiovascular pathologies in regard to H_2_S research, there is a clear overlap with aging-associated pathologies, but also pathologies related to cardiorenal syndrome (CRS) [12,20]. These include heart failure, kidney failure, ischemic events, cardiomyopathy, hypertension, and diabetes, an overview of which is given in Figure 4. While the definition of CRS has not been fully developed, the current definitions provide a useful framework for an examination of H_2_S [20]. Following Ronco et al.’s classification of CRS, pathologies were grouped in the criteria of acute/chronic and the initial location of the syndrome: heart, kidney, or systemic [18]. By examining the changes found in these diseases in the context of the previously discussed physiological balance of H_2_S and other redox molecules, this section addresses the potential role of the antioxidant aspect of H_2_S.

### 5.1. Cardiac Cardiorenal Syndrome (CRS) Pathologies

A well-known effect of aging in humans is the switch from steady increase to a decrease in diastolic pressure after passing middle age (as opposed to the continuous increase of systolic pressure). This is attributed to increased arterial stiffness and hypertrophy of the heart with preserved end diastolic volume [72]. One of the major mechanisms behind this and other afflictions of the heart is uncoupling of NOS, leading to production of O_2_^−^, where H_2_S has been shown to be restored in the experimental setting using rat models [59]. During aging, the level of H_2_S in heart tissue was reduced in rats [60]. Hearts with lower H_2_S levels have been shown to be more prone to disease as well as having reduced functionality and experience uncoupling [59,73]. In other words, it is possible that H_2_S’s antioxidant capabilities are a key aspect of the cardiac pathologies of aging. Notably, such pathologies tend to fall under the CRS.

Acute pathologies initiating in the heart related to CRS include myocardial infarction, heart transplantation, surgery, and acute heart failure from cardiomyopathy [20]. In general, these fall under ischemic events or maladaptation to the environment. In both cases, ROS signaling and H_2_S have been shown to play a role. The role of ROS in pathology is one of overproduction, in which the adaptive processes turn into damaging ones in the form of oxidative stress [55]. H_2_S mitigates the negative effects to a degree in acute disease, however, it can easily become depleted. The amount of H_2_S reserves is indicative of the severity of the disease and damage, as shown by studies on aged rat hearts. This effect is also observed within aging; when comparing aged hearts, the levels of H_2_S also predicted the severity of the disease at the same (calendar) age [74].

Direct ROS scavenging is not the only potential path for H_2_S to mitigate the damage in acute cardiac problems. Inflammation plays a crucial role in the extent of damage that takes place and is responsible for increased ROS levels. In turn, ROS modifies important signaling pathways such as NFκB and Keap1/Nrf2/, which are also targets for H_2_S modification, as previously discussed. H_2_S reduces inflammation, fibrosis, and hypertrophy induced by these pathways, however, it can become rapidly depleted during prolonged or intense oxidative stress [75,76]. Notably, H_2_S has been shown to play an essential role in autophagy, a major underlying mechanism of cardiac injury [77]. In cardiac autophagy, oxidative stress leads to the degradation of mitochondria, the primary source of that stress, as a form of self-survival [78]. H_2_S has the ability to prevent autophagy in experimental settings, thus reducing the long-term effects of acute damage [78].

The chronic CRS pathology initiating in the heart is primarily chronic heart failure [20]. Oxidative stress has been indicated as a major player in heart failure, with much research focused on the activation of matrix metallopeptidases (MMPs), which downregulates the production of H_2_S [54]. The importance of H_2_S in heart failure is demonstrated by the study of Koning et al. on ‘the fate of sulfate’ [25]. In this study, patients with chronic heart failure were followed and their sulfate levels in blood and urine were measured. Patients showed higher plasma levels of sulfate and lower urinary excretion of sulfate compared to healthy controls [22]. While this research shows that sulfate may be useful as a biomarker, it does beg the question of what the extra sulfate is indicative of, especially considering that higher H_2_S in the tissues of rodent models suggests protection. It may be possible that the reduced sulfate excretion in the urine is indicative of renal malfunction in these patients, as low excretion is also correlated with certain forms of renal disease progression [17]. Reduced sulfate excretion will clearly increase plasma levels. It is also possible that the sulfate found in the plasma might be in the form of polysulfides that are produced from the interactions with free radicals [79].

### 5.2. Renal CRS Pathologies

Acute renal initiators of CRS fall under the broad clinical observation of acute kidney injury (AKI), characterized by the loss of kidney function within one week [80]. AKI can be further classified into groups based on the location of the underlying cause, namely prerenal, renal (intrinsic), and postrenal. Interestingly, H_2_S has been found to play a role in pathologies belonging to each of these categories. AKI has the potential to develop into CKD, with oxidative stress, hypoxia, and fibrosis being implicated in the transition [64]. Furthermore, the kidney signals its distress to the rest of the body and attempts to rectify the situation by reducing vascular resistance through various mechanisms [81]. This activation can have long-term consequences for the kidney, as it can lead to renal inflammation and fibrosis [81]. RAAS activation also leads to more workload on the heart due to increased blood volume and vascular resistance, thus contributing to the development or exacerbation of CRS and related cardiac pathologies [81].

Prerenal AKI is characterized by a sharp decrease in blood flow to the kidneys and is therefore not an initiating event in CRS, but is rather secondary to problems of the heart and vasculature. The most common cause of AKI is prerenal caused by surgery, with cardiac surgery, namely cardiopulmonary bypass, having the highest associations [82,83]. This type of ischemic event is similar to direct renal injury by transplantation, which results in an ischemic/reperfusion event [11,34]. Direct injury by transplantation is a potential initiator of CRS [20]. In both cases, the role of H_2_S as an oxygen sensor and regulator of metabolism is essential [9,84]. At the cellular level, reduced O_2_ leads to reduction in the metabolism of H_2_S, thereby increasing its levels. H_2_S then regulates the mitochondria and oxidative signaling to bring the cell into a lower energetic state [85]. In the case of the renal stem cells, located in the papilla and possibly in the tubules, this brings them into a quiescent state. This protects the tissues from cell depletion in the hypoxic environment, assuming the hypoxia is transient. However, the capacity of endogenous levels of H_2_S to perform this crucial ability is limited and crossing that limit leads to pathology.

Renal AKI also includes various conditions in which the kidney is directly affected [20] such as acute interstitial nephritis [86]. Acute interstitial nephritis is the result of an acute autoimmune reaction or response to an infection (pyelonephritis), or, most commonly, a variety of medications [86,87]. Regardless of the instigator, the cause of disease is inflammation, which in turn can lead to fibrosis in the long-term [87]. Despite having various potent targets for H_2_S, little research has been done on the topic. Chen et al. showed that sepsis induced AKI, a different major inflammatory type of injury, corresponds with H_2_S levels and that renal damage can be ameliorated by introducing more H_2_S [88]. While there are to the best of our knowledge no publications on H_2_S related medication-instigated interstitial nephritis, H_2_S has been shown to improve the safety of nonsteroidal anti-inflammatory drugs (NSAIDs) in other organs [89,90]. NSAIDs are a known potential instigator of interstitial nephritis, while also increasing the chance of developing AKI, particularly in individuals with CKD [91]. On a tangent, acute nephrotoxicity caused by acetaminophen overdose depletes glutathione (and therefore H_2_S) in the kidneys, whilst supplementation with H_2_S donors reduces inflammation and damage [92,93].

Acute tubular necrosis, another cause of intrinsic AKI, can be induced by cisplatin. H_2_S, in turn, has been shown to be protective in cisplatin-induced renal disease [94]. Recent research suggests that the protective effect is a result of SIRT3 modification, leading to attenuation of mitochondrial damage [95]. It is important to note that not all donors create this effect, indeed, the slow releasing H_2_S donor GYY4137 worsens the renal damage by promoting increased oxidative stress [96]. Furthermore, cisplatin is thought to downregulate CSE, leading to reduced levels of H_2_S in renal cells, potentially the mechanism for its nephrotoxicity [94]. Other forms of tubular necrosis also result from ischemia and/or inflammatory processes, again suggesting a potential role for H_2_S signaling.

Postrenal AKI is caused by the blockage of the urinary tract, causing backflow and/or build-up of urine in the kidneys [97]. This blockage can be caused by a variety of problems, from kidney stones to bacterial infection [97]. The causes may not be directly related to the redox control by H_2_S, however, H_2_S plays an important role in mitigating the damage resulting from this type of injury. Rats undergoing unilateral ureteral obstruction suffer fibrosis and loss of function that can be ameliorated by H_2_S treatment [98]. In particular, GYY4137 seems to mitigate fibrosis via the TGF-B1 mediated pathway, which is involved in crosstalk with NFκB and inflammation [98]. Furthermore, H_2_S speeds up recovery time in the rats that recover from the damage [98]. Should the acute obstruction become a chronic one, then TGF-B1 and ANG II are upregulated, causing an epithelial–mesenchymal transition in the kidney [99]. H_2_S mitigates this effect and has been shown to be protective in ANG II related pathologies [23,99].

CKD is characterized by increased levels of oxidative stress and chronic hypoxia [64]. Nrf2 plays an important role in regulating the oxidative stress under physiological circumstances, however, in CKD, the activation seems to be insufficient [64]. This could be due to the complex interactions that can cause or propagate CKD such as hyperglycemia and hypoxia [64]. In any case, the result is hypoxia, fibrosis, inflammation, oxidative stress, and/or anemia [50]. It is clear, however, that in CKD, H_2_S production capacity is reduced in the kidneys as well as the liver [100]. This is mirrored by homocysteine levels being elevated while H_2_S levels are reduced in patients with CKD, suggesting that the transsulfuration pathway is disrupted [101]. H_2_S interactions with the HIF-1α pathway could be important, as HIF-1α protects against hypoxic injury initially and is downregulated in CKD [50]. Prolonged exposure to HIF-1α activation, however, leads to fibrosis of the kidneys [50]. H_2_S’s dose dependent activation or downregulation could prove important to potential treatment. This, together with H_2_S’s ability to alleviate inflammation, fibrosis oxidative stress, and anemia (through increased EPO synthesis) makes it a solid candidate for future research on CKD. For an in-depth review of H_2_S and CKD, we refer to the extensive paper by Dugbartey [50].

### 5.3. Systemic CRS Pathologies

While not a primary disease of the heart or kidney, the various forms of diabetes mellitus (DM) play an important role in the development and severity of various CRS pathologies. ROS is induced and antioxidant pathways downregulated by advanced glycation end products resulting from hyperglycemia [102]. An example of one such pathway is cAMP, which leads to increased ROS in DM type II and is also regulated by H_2_S [67,103]. At the glomerulus, blood is filtered and excretion products including water and small molecules are transferred to Bowman’s capsule for further processing in the tubuli. Specialized cells called podocytes serve a crucial role in this filtration process and are damaged by hyperglycemia, resulting in reduced GFR. This damage and other forms of diabetic nephropathy (DN) can be prevented by H_2_S [104]. Furthermore, the damage caused by hyperglycemic conditions in diabetic kidney disease (DKD) in general can be relieved by H_2_S [96]. H_2_S production is reduced in hyperglycemic conditions due to excessive MMP-9 activity, thus paving the way for oxidative stress to develop [11]. However, H_2_S serves another role and increases insulin sensitivity and cellular glucose uptake, thus potentially tackling the hyperglycemia at its source [105,106]. Altogether, H_2_S should be considered an important topic of research when it comes to the pathology and treatment of diabetic CRS.

Hypertension is an important pathology due in part to its role as a risk factor for various other pathologies, some with significant morbidity and/or mortality rates [107]. It can be categorized as the more common primary hypertension with no directly attributable cause ([108]) or the rarer secondary hypertension, in which an underlying disorder such as renal dysfunction is the culprit [23,107]. An important clinical distinction is resistant hypertension, defined as hypertension that does not respond to combination therapy of the four major antihypertensive drugs [107]. An estimated 9–18% of hypertensive cases are resistant and can be either essential or secondary, indicating a societal need for better understanding and treatment options for these patients [107,109]. As discussed in the previous section, blood pressure is in part controlled by redox signaling [63,110]. Higher levels of ROS signaling predispose to hypertension and CSE knockout mice develop hypertension [9]. The few studies in humans on H_2_S in diabetes and transplantation that also included blood pressure measurements show H_2_S levels to be inversely associated with blood pressure [24]. The animal models in which H_2_S can effectively reduce blood pressure are numerous including ANG II-induced hypertension [23,111], spontaneously hypertensive rats [112], and L-NNA induced hypertension (inhibition of NO synthesis) [26]. When existing antihypertensive drugs such as angiotensin-converting enzyme inhibitors (ACE-inhibitors) are modified by sulfhydrylation, they act as a donor for H_2_S [112]. In the case of sulfhydrylated ACE-inhibitors, their safety is improved as well as having greater potential in treating hypertension than their standard counterparts through direct effects in the vascular tissue [112]. Due to its various mechanisms of antioxidant activity and numerous effects related to blood pressure control, H_2_S could potentially serve as a new treatment for both kidney-induced secondary hypertension as well as essential hypertension.

Atherosclerosis is a common pathology of aging and comes with potentially debilitating consequences, monitored clinically primarily through concentrations of low density lipoprotein (LDL) and total cholesterol in the blood. Much like hypertension, treatment and prevention of this mostly asymptomatic process is essential for promoting cardiovascular health in the long-term. The endogenous production of H_2_S is perhaps a double-edged sword in atherosclerosis, however, as shown in Figure 5 [30]. Endogenously produced H_2_S is connected to atherosclerosis as a CSE knockout shows accelerated atherosclerosis development, rescuable by H_2_S treatment [113]. Preexisting plaques, on the other hand, can develop micro-vessels through CSE/H_2_S mediated angiogenesis, which increases the risk of plaque ruptures [114]. Exogenous treatment has shown more promise on the whole through suggested mechanisms such as dilating the vessels to restore flow, interactions with the Keap1/Nrf2 redox balance, protecting endothelial functionality from cell senescence, bypassing affected vessels through angiogenesis, protecting against (mitochondrial) DNA damage, and possibly foam cell formation via SIRT1 [115]. H_2_S is also important in the liver’s metabolism of lipids, an essential part of atherosclerosis, and oxidized LDL levels are inversely related to H_2_S [30,115]. While more research is needed in this area before its application in humans, tentative optimism can be had for the potential of H_2_S in preventing atherosclerosis as well as treating early stages, but should be considered with more caution for advanced disease.

## 6. Therapeutic Potential in Cardiovascular and Renal Disease

Considering the wide variety of diseases in which H_2_S has been implicated during recent years, it is not surprising that research has also started on exploiting its potential as a therapeutic drug [116]. A major line of study are the therapeutic applications during ischemic/reperfusion events of both the heart and kidneys [34,84,85,117]. Notably, H_2_S treatment leads to amelioration of aging, sclerosis, and fibrosis in these organ systems. Furthermore, existing medications that interact with the renal-cardiovascular system interplay such as NSAIDs and ACE inhibitors are being modified with sulfur to induce H_2_S production. One such modification, ATB-346, has been shown to improve the safety of NSAIDs and is currently at stage II trials [118]. Beyond direct administration, the combination of H_2_S and precursors present in food and produced by the gut biome open the way for diet potential [32,119,120]. Indeed, H_2_S has been recognized as the primary effector of antioxidant behavior in foods such as garlic and onion [32,119,120]. Another lifestyle change, regular aerobic exercise, has been shown to induce H_2_S in vivo [76]. In short, there is not only a wide variety of targets for H_2_S treatment, but also a wide variety of treatment possibilities (Table 2). However, how much of the therapeutic potential for renal and cardiovascular disease is explained by the antioxidant effect of H_2_S?

### 6.1. Treatments Using H_2_S Donors

H_2_S donors are widely used in various recent studies. While NaHS or Na_2_S is convenient for preclinical experiments, sodium thiosulfate (Na_2_S_2_O_3_) is already registered for use in humans as a treatment for cyanide poisoning as well as being used off-label for calciphylaxis [121,122]. NaHS has been promising in the treatment of atherosclerosis in rat models [113] and hypertension in spontaneously hypertensive rats, Dahl salt sensitive rats, ANG II infusion, and NO synthesis inhibition [24]. It has been shown to improve renal function, reduce extent of injury, improve recovery rate after injury, treat diabetic renal disease, and increase tubular regeneration [9,105,123]. NaHS has also been used to ameliorate aging of various tissues including the kidneys and heart [12,65]. Sodium thiosulfate is beneficial in treating ANG II induced hypertension, renal damage, and heart disease [23,26,111,124]. It has also shown promise in rat models of cardiac ischemia [125,126] and preeclampsia [127] as well as mouse models of cardiomyopathy [128]. Sodium thiosulfate has been shown to be safe in intravenous doses up to at least 15 grams in patients with acute coronary syndrome undergoing coronary angiography [129]. In all of these cases, evidence is mounting that the interactions of H_2_S and redox signaling is the main mechanism leading to these benefits [74,76,88,111,130].

### 6.2. Improving Transplantation Success

Along the same lines as its use in the treatment of ischemia/reperfusion events, H_2_S has been shown to improve solid organ transplantation success in rats [131]. This is an important finding considering the limited supply of organs and the role of transplantation to treat end stage kidney or heart disease. As described in Section 4, H_2_S protects tissue from ischemic/reperfusion events by reducing the resulting oxidative stress [84,85]. From a therapeutic standpoint, this would mean that administering H_2_S or upregulating its production could reduce the damage of infarctions, ischemic events, and transplantation [116]. An important point to consider is that H_2_S can be used in the perfusion of donor organs, and when doing so, it induces a hypometabolic state. This state can be used to replace the normal cooling method for transporting the organ, reducing the damage caused by the cooling on top of its benefits regarding oxidative stress. Beyond the success rate of the transplantation, it would be valuable if future research could investigate if H_2_S reduces the prerenal kidney injuries that come with transplantation of other organs, in particular heart transplants, as these can lead to the development of CRS (see Section 5.2).

### 6.3. Diet and Exercise

Recently, H_2_S has been shown to be involved in the benefits exercise and diet can have on cardiovascular health. In the case of exercise, it has been shown that long-term regular aerobic exercise is beneficial in attenuating age-related fibrosis of the heart. Recently, the mechanism of this attenuation was related to H_2_S levels [76]. In the case of diet, the benefits of the allium genus (onions), in particular allium sativum (garlic), has been connected to H_2_S released from diallyl disulfide [50,58]. Diallyl sulfide has been shown to have CSE-dependent benefits related to HIF-1a expression in cellular hypoxia responses [50,58]. Many of the benefits of these foods have been attributed to their antioxidant properties [32,119,120]. As H_2_S’s protective effects depend upon activation of endogenous antioxidant capacity, it may surpass the disappointing results that direct antioxidant treatment have shown thus far [132].

### 6.4. Regenerative Medicine

Stem cell treatment has shown promise in treating serious heart disease including heart failure [133]. There are several proposed mechanisms of this benefit including differentiation, immunomodulatory factor, and H_2_S secretion [134]. However, its effectiveness can vary greatly between individuals. Indeed, many clinical trials on stem cell treatment focus on using stem cells obtained from the patient, which can be of low quality considering the condition of these patients [133]. The quality of stem cells and their ability to differentiate requires a minimum level of ROS, however, high levels lead to oxidative stress that results in senescence or death of the stem cells [135]. This can be particularly challenging due to simple matters that might take place in a treatment protocol such as exposure to air, resulting in an increase of ROS in readily available stem cell sources such as mesenchymal stem cells (MSCs) [134]. Considering the previously discussed ways that H_2_S antagonizes ROS, it stands to reason that H_2_S co-therapy might create a synergistic effect by better enabling stem cells to reach greater therapeutic potential while at the same time providing protection on its own. Indeed initial research in bone indicates that H_2_S does indeed preserve MSC function [135]. Recently, Abdelmonem et al. found that preconditioning in vitro or co-delivery in vivo using NaHS both resulted in improved outcomes in treating rats with heart failure over MSC treatment alone [136]. Any combination of NaHS and/or MSCs resulted in no notable fibrosis. All combinations improved ejection fraction, fractional shortening, and the left ventricle diameter in heart failure with the preconditioned MSC treatment coming closest to healthy controls. Furthermore, preconditioned MSCs were able to restore QRS duration and QT intervals to healthy control levels, whereas NaHS alone or in combination with MSCs did not [136]. While still in an early phase, combination therapy of stem cells and H_2_S has the potential to become a novel therapy.

## 7. Discussion

All in all, H_2_S research has thus far opened new avenues of research into important diseases of the cardiovascular and renal systems, many of which need more or better treatment options. A great deal of H_2_S’s potential lies in its antioxidant properties. H_2_S’s ability to activate endogenous antioxidant production, modify key signaling proteins that are also targeted by ROS, and regulate the metabolism of mitochondria make it a complex and interesting puzzle. This complexity may also underlie the benefits seen in the treatment of rodent models of disease and the observations of H_2_S associations with different aspects of various diseases in humans.

Although most of the research has been done in cell lines and in rodent models, we are reaching the point in which translation to the clinic is underway. The existing research in humans has initially mostly been observational, exploring changes in sulfate under various circumstances. Currently a handful of drugs modified to be H_2_S donors are in varying phases of clinical trials. While the focus of these trials is on improving the safety profile of the original drug, the safety data in humans may ease the way for trials aimed specifically on the use of H_2_S donors as a treatment. Sodium thiosulfate is also in various clinical trials, for example, it has proven safe in phase II trials for treatment of acute coronary syndrome (ClinicalTrials.gov identifier NCT03017963) and is undergoing phase III trials. Hopefully, the results in animal models to directly treat pathologies such as those belonging to CRS as well as indirectly by improving other treatment options such as transplantation and stem cell therapies can be replicated in humans.

It is also important to realize that while H_2_S is reduced in cardiovascular and renal pathologies, H_2_S and the transsulfuration enzymes are increased in other pathologies such as cancer and certain genetic neurological disorders [38]. When considering its use in humans, one should be careful to take this into consideration, as it could very well have implications for use in some subpopulations. From the other perspective of treating those pathologies with overproduction, care must also be taken to avoid making the heart and kidneys more vulnerable to oxidative stress related pathologies, exacerbating the already growing CRS problem. A diagram depicting dose-response relationships of H_2_S can be found in Figure 6. However, even if such problems exist, they are not insurmountable, and we remain optimistic about H_2_S’s therapeutic potential.

## 8. Conclusions

H_2_S is a gaseous signaling molecule that plays an important role in redox signaling. Research has exploded on its role in a broad spectrum of biological processes, both physiological and pathological. In the case of the renal and cardiovascular systems, H_2_S plays an important role in maintaining ROS signaling at safe levels by promoting scavenging of ROS as well as competitively modifying cysteine residues on key signaling molecules. As such, depletion of H_2_S is implicated in a variety of age-related pathologies as well as pathologies that fall under CRS. A number of these pathologies are difficult to treat and require novel therapies. Current research suggests potential for H_2_S-based therapies, however, this has been limited primarily to studies in rodents. Fortunately, one donor of H_2_S, sodium thiosulfate, is already registered for use in humans, thus easing the way for translational studies. Furthermore, H_2_S shows potential for improving other forms of treatment such as the safety of NSAIDs, transplantation success, and stem cell therapies. Considering all these points, H_2_S is a prime target for further research with potentially a large clinical impact.

## Figures and Tables

**Figure 1 antioxidants-10-00373-f001:**
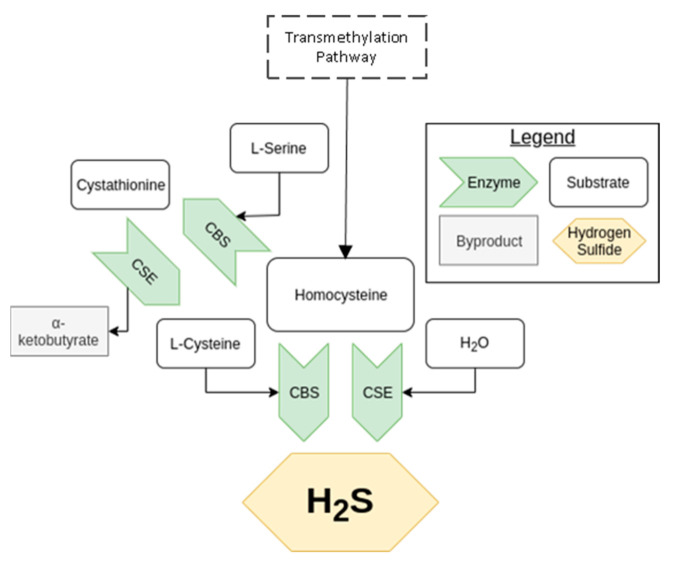
An overview of the transsulfuration pathway. The arrow, representing an enzyme, points toward the product of the reaction it catalyzes.

**Figure 2 antioxidants-10-00373-f002:**
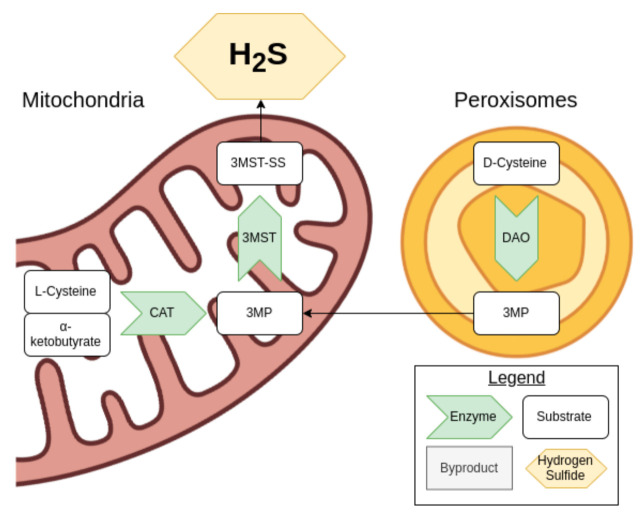
An overview of the cysteine catabolism pathway.

**Figure 3 antioxidants-10-00373-f003:**
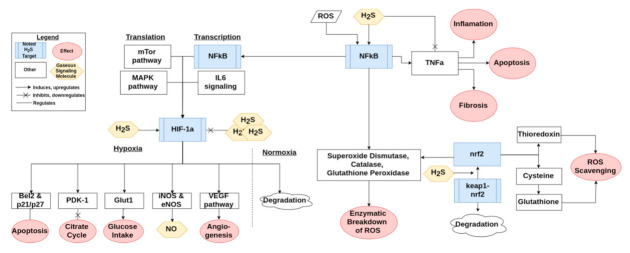
An overview of how H_2_S interacts with the Keap1/Nrf2, NFκb, and HIF-1α pathways. The conglomerate of three H_2_S indicates supraphysiological levels of H_2_S.

**Figure 4 antioxidants-10-00373-f004:**
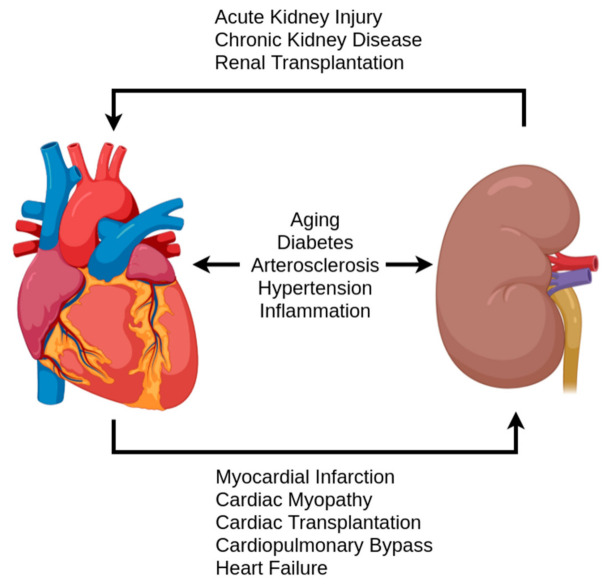
A simplified overview of pathologies and events contributing to the cardiorenal syndrome.

**Figure 5 antioxidants-10-00373-f005:**
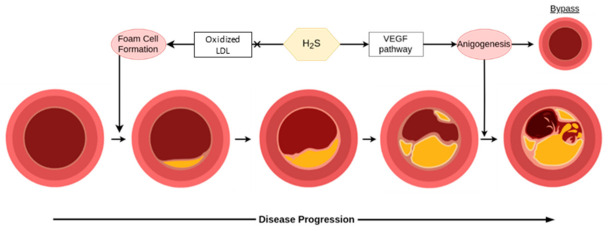
H_2_S is helpful in early stages for the prevention of disease, however, it can be detrimental in later stages of atherosclerosis. In the figure, LDL is low density lipoprotein, and VEGF is vascular endothelial growth factor.

**Figure 6 antioxidants-10-00373-f006:**
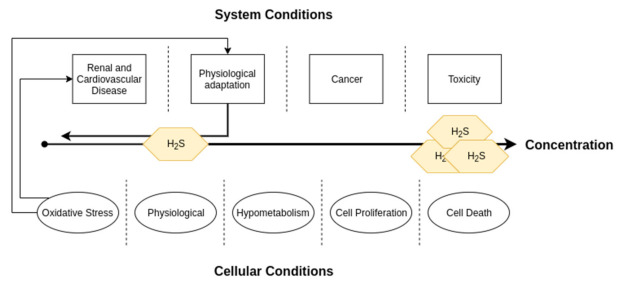
A diagram depicting dose-response relationships of H_2_S concentrations. This review focused on the renal and cardiovascular systems, in which levels of H_2_S lower than physiological amounts can lead to disease. However, it is important to note that high levels are characteristic of cancer and can also lead to toxicity.

**Table 1 antioxidants-10-00373-t001:** A brief overview of the toxic effects of hydrogen sulfide gas exposure.

Concentration	20–50 ppm	100–200 ppm	250–500 ppm	500+ ppm	1000+ ppm
**Effects**	Kerato- conjunctivitis, Airway agitation	Olfactory paralysis (smell disappears), Eye and airway agitation becomes severe	Lung edema that worsens with longer exposure time	Serious eye damage within 30 min Unconscious or dead within 8 h Amnesia	Immediate collapse due to respiratory failure

**Table 2 antioxidants-10-00373-t002:** An overview of some of the potential treatments using hydrogen sulfide.

Administration Type	Examples	Considerations
**Donors**	NaHS, Na_2_S	Easy to use in vitro and in animal models. Phase 1 clinical trials.
	Sodium Thiosulfate (Na_2_S_2_O_3_)	Already used in the clinic, increasingly tested in animal models.
**Modified Drugs**	ACE-inhibitors (Zofenopril, Captopril)	Improved safety and efficacy.
	NSAIDS (Diclonefac, Ibuprofen, Naproxen)	Improved safety.
**Gas**	H_2_S	Non-oral administration, potentially dangerous.
**Modified Treatments**	Transplantation	Improvements over cold preservation with greater success rates.
	Stem Cell Treatments	Protection from oxidative stress during processing. Better treatment results.

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
