# Peer review of "Fighting Oxidative Stress with Sulfur: Hydrogen Sulfide in the Renal and Cardiovascular Systems"

_antioxidants, 2021, doi:10.3390/antiox10030373_

Round 1
Reviewer 1 Report
The review addresses an important topic and collects a lot of consistent data from literature.
Anyway, I suggest the authors to revise the structure of the review, following the instruction of Antioxidants. in particular I suggest to the authors to pay particular attention to the following point:
- At line 41 there is a point 2. Materials and Methods, clearly not part of the review text.
- It would be appropriate to refer to figure 2 in paragraph 2.2 rather than mentioning it at line 84
- Figure 3 is simply the union of figure 1 and figure 2. Among other things in the text there is no particular reference to this figure if not at line 85. For this reason I suggest the authors to eliminate this figure and to add another one reporting the non-enzymatic pathway.
- See the text at line 278: the authors report the abbreviation of (CRS) without the intact test.
- English is not always clear and fluent. Therefore, instead of recommending a substantial revision, I would encourage the authors to be assisted by a scientific editor whose help would certainly increase the readability, thus comprehensibility of the manuscript.
Author Response
Thank you for taking the time to peer review our paper. We have provided a point by point reply below.
- At line 41 there is a point 2. Materials and Methods, clearly not part of the review text.
Thank you for pointing this out. This has been corrected.
- It would be appropriate to refer to figure 2 in paragraph 2.2 rather than mentioning it at line 84
Corrected.
- Figure 3 is simply the union of figure 1 and figure 2. Among other things in the text there is no particular reference to this figure if not at line 85. For this reason I suggest the authors to eliminate this figure and to add another one reporting the non-enzymatic pathway.
The figure has been eliminated, and the text around line 85 altered to better connect with the following paragraphs while excluding the reference to previous figure 3.
- See the text at line 278: the authors report the abbreviation of (CRS) without the intact test.
Corrected.
- English is not always clear and fluent. Therefore, instead of recommending a substantial revision, I would encourage the authors to be assisted by a scientific editor whose help would certainly increase the readability, thus comprehensibility of the manuscript.
We will take your recommendation into consideration and request help from MDPI.
Reviewer 2 Report
Peer Review :
Fighting Oxidative Stress with Sulfur. Hydrogen Sulfide in the Renal & Cardiovascular Systems by Joshua J. Scammahorn et al
This is a review of the effects of hydrogen sulfide on the renal and cardiovascular systems with particular emphasis on the cardiorenal syndrome (CRS). It discusses the effects of hydrogen sulfide on the redox status as directly effects the levels or reactive oxygen species (ROS) and indirectly through the various signaling pathways. It also pays some attention the role of hydrogen sulfide in preserving transplant kidneys and stem cell survival. It also calls attention to the similarity between CRS and other diseases such as diabetes, hypertension atherosclerosis and ageing (if one can call ageing a disease).
This review is fairly short but contains very good illustrations in its figures of some of the processes involved. It brings needed attention to the wider aspects of hydrogen sulfide metabolism and signaling to the clinical problem of CRS and briefly mentions the interrelations between nitric oxide and hydrogen sulfide. They have included an interesting discussion on the maintenance of redox balance recognizing that some amount of ROS is required for proper cellular functioning. The authors discuss the evidence for involvement of ROS in both acute and chronic heart failure and renal disease and the effects of hydrogen sulfide on the mitigation of these diseases. The authors discuss the hypometabolism induced by hydrogen sulfide in the kidney (evidently from the authors laboratory) but do not mention the observed role of hydrogen sulfide in hibernating animals. The discussion of therapeutic potential of hydrogen sulfide and its donors and especially the place of thiosulfate in therapeutic interventions is also brief but interesting and informative. Overall this is good review and suitable for publication.
I have some specific line-by-line comments/corrections
Line 62 perhaps the role of the sympathetic nervous system (SNS) ought to be mentioned as part of the CRS since on line 216-217 there is an example of this involvement . I appears to me that the majority of workers in CRS agree that CRS is a neurohumeral process involving both the RAAS and the SNS.
Fig. 1 homocysteine appears in a box without any origin. Homocysteine is produced by the transmethylation of methionine and perhaps a box with methionine with an arrow to homocysteine would be helpful
Line 148 “….from their –SH residue.” A reference here would be helpful
Line 194-197 ref (54) also shows that sulfide can directly reduce complex IV without going through complex III.
Line 208 “….between NO and ROS.” A reference here would be helpful
Line 210 “…of various proteins.” A reference here would be helpful
Line 219 instead of “sulfated” this should be sulfurated
Line 221 “….which residues are oxidized.” A reference here would be helpful
Line 241 “and apoptosis.” A reference her would be helpful.
Line 307 insert “it” between however and can
Line 307 “reserves” needs to be defined or a reference to “reserves” needs to be made
Line 371 insert “and” between damage and can
Line 377 Should “sulfate” be replaced by sulfide?
Line 405 “…H2S in CKD” reads better as “…in CKD H2S…”
Line 417 Perhaps inserting primary to read “While not a primary disease..” is better
Line 503 “…. ANG II infusion, NO synthesis…” Insert and after infusion and delete the comma
This a good paper and should be published after some minor changes as suggested.
Author Response
- This is a review of the effects of hydrogen sulfide on the renal and cardiovascular systems with particular emphasis on the cardiorenal syndrome (CRS). It discusses the effects of hydrogen sulfide on the redox status as directly effects the levels or reactive oxygen species (ROS) and indirectly through the various signaling pathways. It also pays some attention the role of hydrogen sulfide in preserving transplant kidneys and stem cell survival. It also calls attention to the similarity between CRS and other diseases such as diabetes, hypertension atherosclerosis and ageing (if one can call ageing a disease). This review is fairly short but contains very good illustrations in its figures of some of the processes involved. It brings needed attention to the wider aspects of hydrogen sulfide metabolism and signaling to the clinical problem of CRS and briefly mentions the interrelations between nitric oxide and hydrogen sulfide. They have included an interesting discussion on the maintenance of redox balance recognizing that some amount of ROS is required for proper cellular functioning. The authors discuss the evidence for involvement of ROS in both acute and chronic heart failure and renal disease and the effects of hydrogen sulfide on the mitigation of these diseases. The authors discuss the hypometabolism induced by hydrogen sulfide in the kidney (evidently from the authors laboratory) but do not mention the observed role of hydrogen sulfide in hibernating animals. The discussion of therapeutic potential of hydrogen sulfide and its donors and especially the place of thiosulfate in therapeutic interventions is also brief but interesting and informative. Overall this is good review and suitable for publication.
Thank you for your kind words. Our goal was indeed to provide a short, to the point review.
On the topic of hypometabolism, H2S and hibernation, we indeed did not look into that role in this current paper as we attempt to bring the focus quickly on its potential in humans. This in order to serve as a brief introduction to what H2S is for unfamiliar readers and its potential as a therapeutic solution for disease. We do not feel that including this information adds to this purpose.
- I have some specific line-by-line comments/corrections
- Line 62 perhaps the role of the sympathetic nervous system (SNS) ought to be mentioned as part of the CRS since on line 216-217 there is an example of this involvement . I appears to me that the majority of workers in CRS agree that CRS is a neurohumeral process involving both the RAAS and the SNS.
A very good point. We have added a line to include this.
- Fig. 1 homocysteine appears in a box without any origin. Homocysteine is produced by the transmethylation of methionine and perhaps a box with methionine with an arrow to homocysteine would be helpful
We have adapted the figure to represent that homocysteine is a product of the transmethylation pathway.
- Line 148 “….from their –SH residue.” A reference here would be helpful
We have added existing reference 33 here from line 144.
- Line 194-197 ref (54) also shows that sulfide can directly reduce complex IV without going through complex III.
This interaction has now been added to this segment. Thank you
- Line 208 “….between NO and ROS.” A reference here would be helpful
Reference 56 has now been added here.
Line 210 “…of various proteins.” A reference here would be helpful
Reference 9 and 29 have now been added to the sentence previous to this one. Reference 56, 9 and 29 added to this line.
- Line 219 instead of “sulfated” this should be sulfurated
Thank you, corrected
- Line 221 “….which residues are oxidized.” A reference here would be helpful
New reference added, “Foster DB, Van Eyk JE, Marbán E, O'Rourke B. Redox signaling and protein phosphorylation in mitochondria: progress and prospects. J Bioenerg Biomembr. 2009 Apr;41(2):159-68. doi: 10.1007/s10863-009-9217-7. PMID: 19440831; PMCID: PMC2921908.”
- Line 241 “and apoptosis.” A reference her would be helpful.
Existing reference 55 has been added here.
- Line 307 insert “it” between however and can
Done
- Line 307 “reserves” needs to be defined or a reference to “reserves” needs to be made
Done, defined as H2S reserves.
- Line 371 insert “and” between damage and can
Sentence altered to ‘…and that renal damage can be…’ to better convey the intended meaning.
- Line 377 Should “sulfate” be replaced by sulfide?
Sentence altered to ‘…depletes glutathione (and therefore hydrogen sulfide) in the kidneys whilst…’ to better reflect the intended meaning and address this point.
- Line 405 “…H2S in CKD” reads better as “…in CKD H2S…”
Corrected
- Line 417 Perhaps inserting primary to read “While not a primary disease..” is better
Agreed, done.
- Line 503 “…. ANG II infusion, NO synthesis…” Insert and after infusion and delete the comma
Done